# META DOMAIN REWEIGHTING FOR PARTIALLY KNOWN OUT-OF-DISTRIBUTION GENERALIZATION

## ABSTRACT

Distribution shift poses a significant challenge for modern machine learning methods when applied to real-world scenarios. Existing researches typically assume that either unlabeled data from the target domain is available (domain adaptation) or nothing is known about the target dataset (domain generalization). However, distribution shifts are often caused by environmental changes or human intervention, and some facets of the distribution shift can be predicted while others remain unknown. To address this issue of partial knowledge in out-of-distribution generalization, this paper proposes a model-agnostic reweighting method named *Meta Domain Reweighting for Partially Known Out-of-Distribution Generalization* (PKOOD). Specifically, we utilize a bilevel meta-learning framework to simulate the known distribution shift and automatically determine an effective reweighting of the training samples to achieve strong generalization performance on unknown test datasets. Additionally, we derive the upper bound of the risk gap between the reweighted training samples and the target dataset theoretically and incorporate it as a regularizer to guide loss design for reducing the variance and bias of both known and unknown distribution shifts. The proposed method is evaluated on a real-world people income prediction dataset Adult and a recent out-of-distribution image classification benchmark NICO++, demonstrating its superiority over state-of-the-art algorithms regarding partially known OOD generalization performance.

## 1 INTRODUCTION

Modern machine learning techniques have achieved notable advancements in diverse fields like computer vision and recommender systems. Despite their great success, these methods typically assume that the training data and the test data are independent and identically distributed, known as the i.i.d assumption. In actuality, this assumption may not hold in several real-world situations due to distribution shifts between the training and test data. As a result, utilizing the trained model directly on the new test dataset would lead to performance degradation.

The domain adaptation (DA) setting Farahani et al. (2021) relaxes the i.i.d.assumption and assumes that not only the training data but also the unlabeled data from the test domain, is available during the training process. This setting allows the transfer of knowledge from the training domain to the test domain for reducing the generalization gap between them. For example, importance sampling-based methods Fang et al. (2020); Huang et al. (2006) exploit the unlabeled test data and estimate the density ratio between training and testing distributions directly to debias them. However, these models have limitations as target domain data may not always be accessible in real-world applications. For example, new arrival users in online services cannot be observed. To address this issue, out-of-distribution (OOD) generalization Shen et al. (2021), also known as domain generalization (DG), has been proposed to enable a model to generalize to new test domains. For instance, the Invariant Risk Minimization (IRM) methods Arjovsky et al. (2020); Chang et al. (2020); Zhou et al. (2022b) assume that an invariant representation exists that can predict the target label well across all domains. The invariant predictor can extract some domain-agnostic representation that is also universal to the unknown target data. Distributionally Robust Optimization (DRO) Sagawa et al. (2020) optimizes the worst-case performance over local distribution sets which are generated from the origin training distribution within a tiny distance. Moreover, another line of approach is reweighting, such as Stable Learning Zhang et al. (2021); Shen et al. (2020) and Meta-Learning Zhou et al. (2022a); Li et al. (2018a), which first reweights the data distribution by some assumptions to discard the spurious

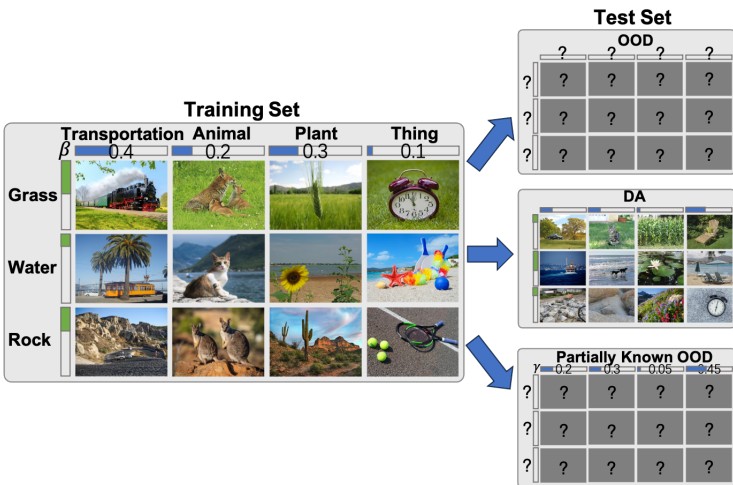

Figure 1: The problem of PKOOD on the NICO++ dataset. The known domains are based on main classes, including transportation, animal, plant and thing. The unknown domains are based on contextual features, such as grass, water and rock. The crucial distinction among the three problems lies in the known information on the test set during training.

features and then performs empirical risk minimization (ERM) on the reweighted distribution. Stable learning aims to find a weight that decorrelates all features for discarding the irrelevant features, while the meta-learning-based reweighting methods use a two-loop way to automatically learn the weight to extract domain-invariant features over multiple source domains.

However, typical out-of-distribution generalization methods assume that the distribution shift is completely unobserved. In reality, the distribution shifts are often caused by environmental changes or human intervention. Priors related to the causal factors of these changes can be acquired to improve out-of-distribution generalization. For instance, in budget-constrained e-commerce recommender systems Anava et al. (2015) on the period of the beginning of the promotion, training data are typically collected before the promotion, while target performance needs to be optimized during the promotion. Although it is impossible to foreknow shifts in user distribution before and during the promotion, changes in the human-presupposed budget distribution can be anticipated. Another example is the prediction of COVID-19 transmission. When using the past spread of COVID-19 in one country to predict the future spread of a new country, we can only know the proportion of COVID-19 variants Callaway et al. (2022) in the two countries, rather than the target data. The crucial distinction among the OOD, DA and PKOOD is shown in the figure 1. To exploit the known test domain distribution, a straightforward way is to reweight or importance sample Fang et al. (2020) the training distribution into the target domain distribution. Nonetheless, this strategy may augment the variance of the learning model, given that the reweighted data deviates from the original training data (as demonstrated mathematically in the following section). Furthermore, even when some aspects of domain distribution are known, unobserved test samples could still introduce unknown domain shifts. Thus, it is crucial to effectively manage the partially known distribution shifts while simultaneously addressing the intricate unknown distribution shifts.

To tackle these challenges posed by distribution shifts, we propose a novel model called **Meta Domain Reweighting for Partially Known Out-of-Distribution Generalization (PKOOD)**. Firstly, we theoretically derive the upper bound of the risk gap through sample reweighting and prove that an optimal domain weight exists to address the partially known OOD problem. Specifically, we employ a meta-learning approach to automatically learn the sample weights of known domains. In detail, to simulate the known distribution shifts, we train the model on the weighted training samples with learnable domain weights in the inner loop of meta-learning architecture and restrict the sample weight as the target distribution in the outer loop. The meta-optimization is to minimize the loss on the target test distribution while descending the loss on the reweighted train loss to reduce the variance of the learned model. Additionally, we use the derived upper bound of the risk as a part of the training loss to reduce the bias between the risk of training and test distribution. In addition, we introduce invariant learning on the inner loop of meta-learning to address the issue of unknown OOD

shifts. By learning an invariant representation, which is stable with the change of environments, we enable unknown OOD generalization. In summary, our model can exploit the known test domain distribution and handle the complex unknown distribution shifts simultaneously.

It is worthwhile to highlight the following contributions of this paper:

- We investigate the problem of out-of-distribution generalization with partially known distribution shift which is widely prevalent in the industry. To the best of our knowledge, our model is the first work to solve this problem.

- We propose a novel model named Meta Domain Reweighting for Partially Known Out-of-Distribution Generalization (PKOOD) based on the reweighting way. We theoretically prove that an optimal domain weight exists to address the partially known OOD problem. Based on the theoretical upper bound of the risk gap, we introduce meta-learning to simulate the known distribution shifts and exploit invariant learning to address the issue of unknown OOD shifts.

- Extensive experimental results demonstrate that the proposed model outperforms baselines on various different datasets under the partially known OOD setting.

## 2 RELATED WORK

### 2.1 INVARIANT RISK MINIMIZATION

**Invariant Risk Minimization** (IRM) Arjovsky et al. (2020) aims at discovering an invariant representation across various environments that can be used to develop an optimal classifier capable of matching all environments. Several works have proposed different variants of the technique to improve performance based on this crucial concept. REx Krueger et al. (2021) utilizes the variance of risks from different environments as a regularizer to achieve an equivalent risk and an invariant classifier. Nonetheless, several works Lin et al. (2021); Zhou et al. (2022b) have noted that IRM-based methods are less effective when applied to overparameterized deep neural networks due to overfitting. SparseIRM Zhou et al. (2022b) proposes adding a sparsity constraint to reduce the number of parameters and, consequently, avoid overfitting to tackle this issue.

### 2.2 DISTRIBUTIONALLY ROBUST OPTIMIZATION

**Distributionally Robust Optimization** (DRO) Levy et al. (2020); Sagawa et al. (2020) is an approach that optimizes the worst-case performance over a set of uncertainty distributions. For example, f-divergence DRO Duchi & Namkoong (2018) and Wasserstein DRO Shafieezadeh Abadeh et al. (2015) limit the distributional shift within the uncertainty set, constrained by f-divergence and Wasserstein distance, respectively. CVar DRO Levy et al. (2020) proposes to optimize the distributionally robust optimization of convex losses with conditional value at risk (CVaR) uncertainty sets. Besides, GroupDRO Sagawa et al. (2020), which follows the multi-domain setting, minimizes the worst-case training loss over a set of pre-defined groups instead of uncertainty sets.

### 2.3 REWEIGHTING

Sample or domain reweighting Li et al. (2018b); Fang et al. (2020) are classic methods used to tackle distribution shifts. These methods typically involve reweighing the samples and training on the weighted ERM. For instance, importance sampling-based methods Fang et al. (2020); Huang et al. (2006) estimate the density ratio between training and testing distributions by leveraging the unlabeled test data, and subsequently reweight the training data by importance sampling to align with the target sampling distributions. However, the availability of unlabeled test data cannot always be guaranteed. Here we introduce two kinds of classic reweighting methods on out-of-distribution generalization, stable learning, and meta-learning.

**Stable Learning.** Stable learning Xu et al. (2021) proposes learning sample weights to ensure statistically independent features in the reweighted distributions. For instance, StableNet Zhang et al. (2021) extends this approach into non-linear deep models by Random Fourier Features to decorrelate features and eliminates statistical correlations between relevant and irrelevant features.

**Meta Learning.** Several researchers Zhou et al. (2022a); Li et al. (2018a); Shu et al. (2019) have recently proposed using meta-learning to address the OOD problem. In this approach, the training data is split into support sets (meta-train sets) and query sets (meta-test sets), and distribution shifts are simulated to generate a model that is robust to such shifts. Using the Model-Agnostic Meta-Learning (MAML) Finn et al. (2017) framework, MLDG Li et al. (2018a) randomly selects a domain as the meta-test set and other domains as meta-train sets to simulate domain shift within mini-batches.

While all of these methods assume no prior knowledge about the test distribution, in real-world industry scenarios, we may have some information about the trend or domain distribution in the test data, even if we cannot obtain the entire unlabeled test data. To address this challenge of partially known OOD generalization, we propose a model that leverages this prior knowledge.

## 3 PRELIMINARIES

### 3.1 NOTATIONS AND DEFINITION

Let $\mathcal{X} \subset \mathbb{R}^{\mathcal{D}}$ denotes the input features and $\mathcal{Y} \in \{0, 1\}$ denotes the label. The domain $\mathcal{D}_\alpha$ corresponds to a probability distribution over $\mathcal{X}$. Our goal is to find a hypothesis $h : \mathcal{X} \to \{0, 1\}$, that minimizes the risk between $h(x)$ and $y$ for a given sample $(x, y)$ where $x \in \mathcal{X}, y \in \mathcal{Y}$. The risk on domain $\mathcal{D}_\alpha$ is defined as $\epsilon_\alpha(h) = \mathrm{E}_{x \sim \mathcal{D}_\alpha}[\mathcal{L}(h(x), y)]$ where the loss $\mathcal{L} : \mathcal{Y} \times \mathcal{Y} \to R_+$ quantifies the difference between $h(x)$ and the true label $y$. We consider an out-of-distribution generalization setting with a source domain $\mathcal{D}_s$ for the training dataset and a target domain $\mathcal{D}_t$ for the test dataset. Our partially OOD setting involves both known distributions $\mathcal{G}_1$ and unknown distributions $\mathcal{G}_2$. Following GroupDRO Sagawa et al. (2020), the two distributions are discretized using different group annotations, denoted by $\mathcal{G}_1 \in \{1, 2, ..., N\}$ and $\mathcal{G}_2 \in \{1, 2, ..., M\}$. For example, in the Figure 1, the known domains $\mathcal{G}_1 = \{Transportation, Animal, Piant, Thing\}$ and the unknown distribution $\mathcal{G}_2 = \{Grass, Water, Rock\}$. A training sample $v$ is denoted as $(x_v, g_{1v}, g_{2v})$, where $x_v$ represents the features of sample $v$, $g_{1v} \in \mathcal{G}_1$ is the known group annotation, and $g_{2v} \in \mathcal{G}_2$ is the unknown group annotation. In the case of test samples, $g_1$ and $g_2$ are unobtainable. Splitting by the known domains, the training distribution $\mathcal{D}_s$ is represented as a mixture of the $N$ known domain distributions with mixing weights $\beta$, i.e., $\mathcal{D}_s = \sum_{i=1}^{N} \beta_i \mathcal{D}_i$. Here $\mathcal{D}_i = p(x|g_1 = i)$ is the sample distribution of a given known domain $g_1 = i$, and $\beta_i = p(g_1 = i) = \frac{\sum_v I(g_{1v}=i)}{\sum_{i=1}^N \sum_v I(g_{1v}=i)}$ is the distribution of known domains. $I(\cdot)$ is the indicator function. Then the risk of training data is $\epsilon_s = \sum_i \beta_i \epsilon_i(h)$, where $\epsilon_i(h)$ is the risk on the known distribution $\mathcal{D}_i$. Similarly, the test distribution is $\mathcal{D}_t = \sum_{i=1}^{N} \gamma_i \mathcal{D}_i'$, where $\gamma$ is the test distribution of known domains and $\mathcal{D}_i'$ is the sample distribution of a given known domain $g_1 = i$ on test dataset. The known sub-domain $\mathcal{D}_i'$ differs from $\mathcal{D}_i$ in training distribution due to the unknown distribution shifts that induce inconsistency between the $p_{train}(x|g_1 = i)$ in the training set and $p_{test}(x|g_1 = i)$ in the test set. We define partially known out-of-distribution generalization as follows:

**Definition 1** (Partially Known Out-of-distribution Generalization). *Given the training data $\mathcal{D}_s$, the $\beta$ distribution over known domains $\mathcal{G}_1$ on training data and the distribution $\gamma$ over known domains $\mathcal{G}_1$ on test data, but the test data $\mathcal{D}_t$ is unknown, and the distribution of $\mathcal{D}_s$ and $\mathcal{D}_t$ differ significantly (non-iid). The goal is to train a model $h$ on the train data $\mathcal{D}_s$ to minimize the empirical risk on the unknown test data $\epsilon_t(h)$.*

The crucial distinction among the OOD, DA and PKOOD is depicted in Figure 1. In DA setting, the unlabeled test data is assumed to be observed. In OOD setting, information regarding the test data is completely unknown. While in PKOOD setting, we are able to touch known domain distribution $\gamma$.

### 3.2 THEORETICAL ANALYSIS

In this subsection, we present a theoretical analysis of the upper bound of the target empirical risk $\epsilon_t(h)$. We employ a commonly used reweighting way to solve the OOD problem. Specifically, we generate a dataset $\hat{\mathcal{D}}_\alpha$, by reweighting samples from $\mathcal{D}_s$ based on $\alpha$ distribution over known domains $\mathcal{G}_1$, i.e., $\hat{\mathcal{D}}_\alpha = \sum_{i=1}^{N} \alpha_i \mathcal{D}_i$. The corresponding empirical risk, which is also known as the empirical $\alpha$-weighted error of function $h$, is denoted by $\hat{\epsilon}_\alpha(h) = \sum_{i=1}^{N} \alpha_i \hat{\epsilon}_i(h)$. The ideal $\alpha$-weighted error $\epsilon_\alpha(h)$ is defined analogously. Besides, we provide the spurious target distribution with unknown

distribution unchanged as $\overline{\mathcal{D}}_t = \sum_{i=1}^N \gamma_i \mathcal{D}_i$ and its corresponding empirical risk is denoted as $\overline{\epsilon}_t(h)$. Since the unknown distribution shift, the target distribution $\mathcal{D}_t = \sum_{i=1}^N \gamma_i \mathcal{D}_i'$ is different from $\overline{\mathcal{D}}_t$. Our goal is to minimize target risk $\epsilon_t(h)$. In this regard, we analyze the difference between $\hat{\epsilon}_\alpha(h)$ and $\epsilon_t(h)$ if we use $\hat{\epsilon}_\alpha(h)$ to train our model. By applying the triangle inequality, we obtain an expression for the upper bound:

$$|\hat{\epsilon}_\alpha(h) - \epsilon_t(h)| \le \underbrace{|\hat{\epsilon}_\alpha(h) - \epsilon_\alpha(h)|}_{variance} + \underbrace{|\epsilon_\alpha(h) - \overline{\epsilon}_t(h)|}_{bias_{known}} + \underbrace{|\overline{\epsilon}_t(h) - \epsilon_t(h)|}_{bias_{unknown}}. \qquad (1)$$

The first $variance$ term shows the error due to the difference between sampled $\alpha$ data and the ideal $\alpha$ data. The second $bias_{known}$ term represents the deviation between the ideal $\alpha$ distribution and the mock target test distribution $\overline{\epsilon}_t(h)$ if there is no unknown distribution shift. The third $bias_{unknown}$ term denotes the deviation due to unknown distribution when the known distributions are consistent. We give the upper bound of the three terms as following theorems.

**Theorem 1.** *Let $\mathcal{H}$ be a finite hypothesis space on $\mathcal{X}$ and $|\mathcal{H}|$ be the size of the hypothesis space. For any $\eta \in (0,1)$, with probability at least $1 - \eta$,*

$$|\hat{\epsilon}_\alpha(h) - \epsilon_\alpha(h)| \le \sqrt{\frac{1}{2m} \sum_{i=1}^N \frac{\alpha_i^2}{\beta_i} \ln(\frac{2|\mathcal{H}|}{\eta})}. \qquad (2)$$

**Theorem 2.** *Let $h^* = \arg\min_{h \in \mathcal{H}} \{\epsilon_\alpha(h) + \overline{\epsilon}_t(h)\}$ and $d_{\mathcal{H}\Delta\mathcal{H}}(\mathcal{D}_i, \mathcal{D}_j)$ is $\mathcal{H}$-divergence between domain $\mathcal{D}_i$ and $\mathcal{D}_j$, we have*

$$|\epsilon_\alpha(h) - \overline{\epsilon}_t(h)| \le \epsilon_\alpha(h^*) + \overline{\epsilon}_t(h^*) + \frac{1}{2} \sum_{i=1}^N \sum_{j=i+1}^N |(\alpha_i\gamma_j - \alpha_j\gamma_i)| * d_{\mathcal{H}\Delta\mathcal{H}}(\mathcal{D}_i, \mathcal{D}_j). \qquad (3)$$

**Theorem 3.** *The upper bound of $|\overline{\epsilon}_t(h) - \epsilon_t(h)|$ is irrelevant to $\alpha$ and under the assumption of IRM invariant predictor Chang et al. (2020) over unknown domains $\mathcal{G}_2$, its expectation is zero.*

The proofs of these theorems and the following corollaries are provided in Appendix A.4. By the Theorem 1 and 2, we have the following corollaries.

**Corollary 1.** *For $\forall i \in \{1, ..., N\}, \alpha_i = \beta_i$, the upper bound of $variance$ in Theorem 1 gets minimum value.*

**Corollary 2.** *For $\forall i \in \{1, ..., N\}, \alpha_i = \gamma_i$, the upper bound of $bias_{known}$ in Theorem 2 gets minimum value.*

To summarize, we have the following observations: (1) The $variance$ and $bias_{known}$ result from the distribution shift of the known distribution. Reweighting into train distribution ($\alpha = \beta$) wound get a low variance but a high bias, while reweighting into test distribution ($\alpha = \gamma$) wound obtain a low bias but a high variance. Thus an optimal weight between $\beta$ and $\gamma$ exists to minimize the sum of the two upper bounds. (2) The $bias_{unknown}$ arises from the distribution shift of the unknown distribution. We can minimize the upper bound through invariant representation learning.

Following these theorems and observations, we propose our model Meta Domain Reweighting for Partially Known Out-of-Distribution Generalization (PKOOD).

## 4 META DOMAIN REWEIGHTING FOR PARTIALLY KNOWN OOD

In fact, it is not possible for the previous three upper bounds to be optimal simultaneously, and some of the bounds may not be computable. Nevertheless, we propose a model that is guided by these upper bounds in order to minimize the loss. Our key idea is to utilize a learnable $\alpha$ distribution to reweight the train distribution, reducing the upper bound and achieving optimal risk on the test dataset. Meta-learning approaches aim to enable the model learning to learn. The meta-learning procedure involves applying large support sets and small query sets to train the model. Our goal is to train a model on the training dataset that exhibits good performance on the target test dataset, despite distribution shifts. Therefore, using meta-learning to solve our problem is appropriate. Meta-learning can have two effects in our setting: (1) It can simulate the known distribution shift to reduce the

*variance* and $bias_{known}$, and (2) it can automatically learn the hyper-parameters domain weights $\alpha$ of the trained model, in addition to the parameters of the trained model.

To achieve this, we first follow MAML architecture Finn et al. (2017) and partition the training data into support sets and query sets in each mini-batch. Random selection is used to assign $80\%$ of data as support sets, while the remaining $20\%$ is allocated as query sets. We make modifications to our approach by reweighting the sample of support sets as $\alpha$ distribution and the sample of query sets as $\gamma$ distribution to replicate the known distribution shift that exists in real-world scenarios. By adopting this strategy, we are able to attain good generalization in target test datasets over multiple iterations. The model framework is illustrated in Appendix A.1.

**Meta-train.** In the meta-train stage, the reweight risk $\hat{\epsilon}_\alpha(h)$ is utilized to approximate the target risk $\epsilon_t(h)$. The loss function of the reweighted risk on the support set is given,

$$\mathcal{L}_m = \hat{\epsilon}_\alpha(h) = \sum_i \sum_{x_j \in \mathcal{D}_i} \frac{\alpha_i}{\beta_i} \ell_\Theta(h(x_j, y_j)), \tag{4}$$

where $\ell$ is the loss function between the predicted and true labels, and in the multi-class classification, it is set as a cross-entropy loss. The sample weights are divided by $\beta$ because the training data follows $\beta$ distribution. To further minimize the gap between $\hat{\epsilon}_\alpha(h)$ and $\epsilon_t(h)$, upper bounds from corresponding theorems are added as regularizers. However, since the upper bound of variance in Theorem 1 is a probability inequality and not very tight, thus rendering it unsuitable for model constraints. According to Hastie et al. (2009), increasing random sampling times can help to reduce the variance of model training. Consequently, through the random splitting of support and query sets over many iterations on the bi-level of meta-learning, the variance term can be reduced.

To reduce the $bias_{known}$, we aim to decrease the upper bound in Theorem 2. Following the work of Ben-David et al. (2010), we assume the existence of a hypothesis $h^* = \arg\min_{h \in \mathcal{H}} \{\epsilon_\alpha(h) + \overline{\epsilon}_t(h)\}$ with low error on both the data with reweighted $\alpha$ distribution and the target domain. If such a hypothesis does not exist, the difference between train and test distributions would be too significant, making it impossible for any OOD method to overcome the problem. Therefore the term $\epsilon_\alpha(h^*) + \overline{\epsilon}_t(h^*)$ becomes much smaller than the next term $\frac{1}{2} \sum_{i=1}^{N} \sum_{j=i+1}^{N} |(\alpha_i \gamma_j - \alpha_j \gamma_i)| * d_{\mathcal{H}\Delta\mathcal{H}}(\mathcal{D}_i, \mathcal{D}_j)$ in the upper bound, which is added as a regularizer to reduce the bias. Since the $\mathcal{H}$-divergence can not be calculated directly, its upper bound $2\sup_{h_d \in \mathcal{H}_d} [p(h_d(x) = i) - p(h_d(x) = j)]$ based on Ganin & Lempitsky (2015) is used where the $p(h_d(x) = i)$ is the probability of sample $x$ belongs to domain $\mathcal{D}_i$. This bound is reached by the optimal domain discriminator $h_d$, which can be learned using the domain label. The regularizer of $bias_{known}$ is definded as follows:

$$\mathcal{L}_k = \lambda_1 \sum_{i=1}^{N} \sum_{j=i+1}^{N} |(\alpha_i \gamma_j - \alpha_j \gamma_i)| * \sum_x |p(h_d(x) = i) - p(h_d(x) = j)| + \lambda_2 \sum_i \sum_{x_j \in \mathcal{D}_i} \ell_{\Theta_d}(h_d(x_j), i), \tag{5}$$

where $\ell_{\Theta_d}$ is the loss of domain discriminator such as cross-entropy.

To reduce the $bias_{unknown}$, Theorem 3 suggests using an IRM-based method to learn invariant representations that do not vary with the unknown domains. In reality, any IRM methods can be used here. For simplicity, we use the simple IRMv1 Arjovsky et al. (2020) over the unknown domain $g_2$ as a regularizer:

$$\mathcal{L}_u = \lambda_3 \sum_{g_2 \in \mathcal{G}_2} ||\nabla_{w|w=1.0} \sum_{(x_j, g_2)} \ell_\Theta(w \cdot h(x_j), y_j)||^2. \tag{6}$$

In total, the support loss combining the losses in Equation 4,5,6 is expressed as follows:

$$\mathcal{F}(\alpha, \Theta) = \mathcal{L}_m + \mathcal{L}_k + \mathcal{L}_u. \tag{7}$$

This optimization aims to minimize the risk of the reweighted domain distribution along with the bias. The gradient of support loss is calculated for all parameters except $\alpha$. The new parameters are obtained using gradient descent $\Theta' = \Theta - \lambda \frac{\partial \mathcal{F}(\alpha, \Theta)}{\partial \Theta}$. Instead of updating the model parameters immediately, they are used as input for the next meta-test stage.

**Meta-test.** In the meta-test stage, query sets are reweighted to simulate the target distribution using the $\gamma$ vector. The query loss is then calculated as a weighted sum of losses on the query samples:

$$\mathcal{G}(\Theta') = \sum_i \sum_{x_j \in \mathcal{D}_i} \frac{\gamma_i}{\beta_i} \ell_{\Theta'}(h(x_j, y_j)). \tag{8}$$

**Update of $\alpha$.** Domain weights $\alpha$ is updated based on the query loss. Although the loss equation does not have $\alpha$, $\Theta'$ is a function of both $\alpha$ and $\Theta$. The gradient of query loss with respect to $\alpha$ is calculated as $\frac{\partial \mathcal{G}(\Theta')}{\partial \alpha} = \frac{\partial \mathcal{G}(\Theta')}{\partial \Theta'} \frac{\partial \Theta'}{\partial \alpha}$. Then, $\alpha$ is updated by gradient descent $\alpha_{new} = \alpha - \lambda \frac{\partial \mathcal{G}(\Theta')}{\partial \alpha}$ where $\lambda$ is the learning rate.

**Update of $\Theta$.** After obtaining the new value of $\alpha_{new}$, we update the model parameter $\Theta$. Due to the large number of model parameters in a deep neural network, it may not be sufficient to optimize the model solely based on the query loss on the query sets. As such, we simultaneously optimize the support loss and query loss to aid in the learning of the model as follows:

$$\Theta_{new} = \Theta - \lambda \frac{\partial (\mathcal{G}(\Theta') + \mathcal{F}(\alpha_{new}, \Theta))}{\partial \Theta}. \tag{9}$$

Finally, when the model is converged, we deploy model parameters $\Theta$ to obtain the predicted label $h(x)$ on the target OOD datasets.

## 5 EXPERIMENT

In this section, we present the evaluation of our proposed PKOOD method on two real-world datasets with various baselines.

### 5.1 DATASETS AND BASELINES

To validate our model on the partial out-of-distribution setting, we conduct experiments on two different types of datasets, namely Adult and NICO++. Besides, an online A/B testing in real industry scenario is conducted in the Appendix A.6

**Adult** is a binary classification task to predict if an adult earns more than 50K per year, as described in Yu et al. (2022). The features of race and sex in this dataset are naturally used to produce sub-populations for different environments. Specifically, we create five environments with known distributions $\mathcal{G}_1$ using the race feature and two environments with unknown distributions $\mathcal{G}_2$ using the sex feature. In the case of unknown distribution, we use one environment as the training dataset and others as testing datasets. For known distribution, we resample the dataset to create different distributions by adjusting the proportions of each environment as $\alpha$ and $\beta$. We create $\alpha$ and $\beta$ as the proportions $x : \frac{x+1}{2} : \frac{x+1}{2} : 1 : 1$ with randomly shuffled. Different values of $x$ from $5, 10, 20, 50, 100$ are set, and we calculate the Euclidean distance of $\beta$ and $\gamma$ to determine the degree of distribution shift.

**NICO++** Zhang et al. (2022) is an OOD image classification benchmark that simulates real-world scenarios where there may be arbitrary shifting between the training and test distributions. Images in this dataset are labeled with both main concepts/categories and the contexts in which visual concepts appear. To support the partial OOD setting, we label the categories into four main classes, namely transportation, animal, plant, and thing. We adjust the proportions of the main classes to generate known distributions $\mathcal{G}_1$. The contexts (e.g. on grass and in water) are used to create unknown distributions $\mathcal{G}_2$, ensuring that the context of the testing dataset is different from that of the training dataset.

To demonstrate the superiority of our PKOOD, we compare with several representative state-of-the-art methods, including **ERM**, IRM based(**IRMv1** Arjovsky et al. (2020)), DRO based(**GroupDRO** Sagawa et al. (2020),**CVarDRO** Levy et al. (2020)), Stable Learning(**StableNet** Zhang et al. (2021)) and meta-learning based(**MLDG** Li et al. (2018a), **MAPLE** Zhou et al. (2022a)).

We optimize the objective function using the Adam Kingma & Ba (2014) optimizer with a learning rate of 0.001. To ensure a fair comparison, we uniformly used a two-layer deep neural network for the Adult dataset and pre-trained ResNet He et al. (2016) for the NICO++ dataset in all methods. The embedding size was set to 64, and the batch size was set to 1024. When applying meta-learning-based methods (MLDG, MAPLE, PKOOD), we need to split the batch train data into a support set and a query set. Although random splitting is possible, we used domain annotations to split the data instead to simulate distribution shift. For methods that require demanding domain annotations (IRMv1, GroupDRO, MLDG, MAPLE), we split the domain based on the Cartesian product of known and unknown distribution domain labels.

### 5.2 CASE STUDY

In this subsection, we demonstrate that there exist reweight values between train distribution and test distribution to achieve the best performance. This conclusion is also drawn from the corollaries in the Theoretical Analysis section and motivates our proposed domain reweighting methods. We conduct a case study experiment on the smaller Adult dataset with biggest degree. The reweight method that balances the known train distribution

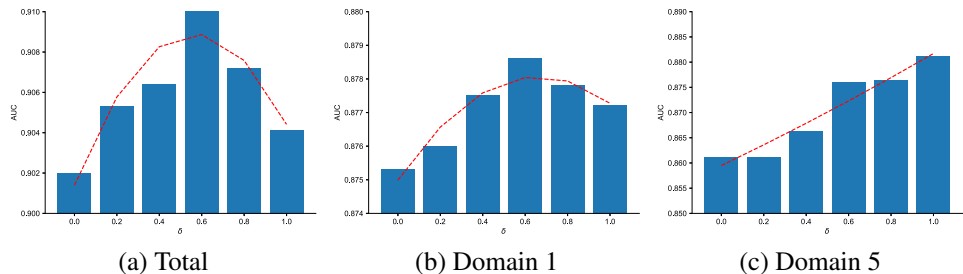

(a) Total        (b) Domain 1        (c) Domain 5

Figure 2: Case: the performance of the Reweight-$\delta$ method on the Adult dataset. The red lines are the trend lines.

Table 1: AUC result in Adult dataset with different degrees of OOD.

| Method | 5 | 10 | 20 | 50 | 100 | Avg. |
|---|---|---|---|---|---|---|
| Distance | 0.4861 | 0.6205 | 0.6964 | 0.7564 | 0.7673 | 0.6653 |
| ERM | 0.8748 | 0.8954 | 0.9083 | 0.8636 | 0.9020 | 0.8889 |
| IRMv1 | 0.8821 | 0.8863 | 0.9102 | 0.852 | 0.8883 | 0.8838 |
| GroupDRO | 0.8869 | 0.9026 | 0.9080 | 0.8642 | 0.9030 | 0.8929 |
| CVarDRO | 0.8765 | 0.8839 | 0.9013 | 0.8625 | 0.9038 | 0.8856 |
| StableNet | 0.8795 | 0.8909 | 0.9222 | 0.8631 | 0.9017 | 0.8915 |
| MLDG | 0.8860 | 0.8962 | 0.9128 | 0.8668 | 0.8962 | 0.8916 |
| MAPLE | 0.8875 | 0.8921 | 0.9191 | 0.8680 | 0.9158 | 0.8965 |
| PKOOD | **0.8880** | **0.9095** | **0.9253** | **0.8803** | **0.9324** | **0.9071** |
| Improve over ERM | 1.51% | 1.57% | 1.87% | 1.93% | 3.37% | 2.05% |

$\beta$ and test distribution $\gamma$ is designed to validate our assumption. For simplicity, we use a simple scalar value $\delta$ to trade-off the two distributions and reweight the samples by $w_\delta = \frac{\delta\beta + (1-\delta)\gamma}{\beta}$. We refer to this method as Reweight-$\delta$. It is easy to observe that Reweight-1 is equivalent to following the train distribution exactly (i.e., ERM), and Reweight-0 is equivalent to reweighting the train dataset into test distribution. The results of Reweight-$\delta$ methods, where $\delta$ is changed from $\{0, 0.2, 0.4, 0.6, 0.8, 1.0\}$, are shown in Figure 2. Figure 2 (a) shows the total performance over all domains. We can observe that the performance initially raises as $\delta$ increases and then decreases later. The poorer performance of Reweight-1(ERM) indicates the presence of a distribution shift problem. The AUC curve is non-monotonous, and neither Reweight-1 nor Reweight-0 achieve the best performance. This is understandable because Reweight-0 has a larger variance and lower bias, and Reweight-1 has a larger bias and lower variance. Therefore, an optimal $\delta$ exists that minimize the sum of bias and variance, i.e., the inflection point in the figure. This result is consistent with the deduction of the previous corollaries. Figure 2 (b) and (c) show the performance on two different domains. The best performances on different domains vary (0.6 in domain 0 and 1.0 in domain 5), indicating that a scalar $\delta$ can not achieve the best performance on all domains. Therefore, we apply meta-learning to learn self-adapting domain weights.

## 5.3 OVERALL PERFORMANCE COMPARISON

We evaluate our method over all the baselines in two datasets, Adult and NICO++.

**Results on Adult dataset.** In this dataset, the task involves binary classification. To evaluate the classification performance, we utilize the area under the curve (AUC) metric Fawcett (2006) In Table 1, we present the results of all methods in the different degrees of out-of-distribution (OOD) settings, as well as their average results. The

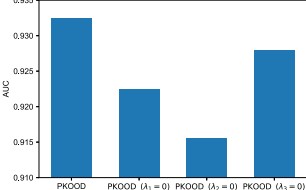 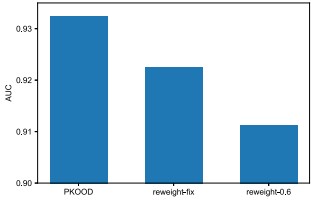

Figure 3: Compared to the variants $\lambda_1 = 0$, $\lambda_2 = 0$, $\lambda_3 = 0$ (Left); Reweight-fix (Right).

degree is defined in the previous subsection. It is worth noting that since the datasets across different degrees are from the distinct random splitting of train and test data and various domain permutations, the results are not comparable across different degrees. It is easy to see that our proposed PKOOD outperforms the baselines in all datasets, showcasing its effectiveness. Additionally, the empirical risk minimization (ERM) method produces the worst results, highlighting the importance of addressing the OOD problem. As the distance between known distribution shifts increases, our improvement over ERM becomes more apparent, indicating its increased robustness to the more severe OOD problem. Moreover, the meta-learning based methods (MLDG, MAPLE) exhibit better performance than other baselines, which verifies the effectiveness of the bilevel approach to simulate distribution shift. Furthermore, our models outperform the meta-learning based methods by considering the known distribution shift by simulating the distribution shift and restricting the theoretical upper bound of bias.

**Results on NICO++ dataset.** The NICO++ dataset is a multi-class classification task. To evaluate the results, we utilize the TopK accuracy metric with K values of 1, 3, 5, 10. The results are shown in Table 2. In terms of performance, the outcomes are comparable to those of the Adult dataset, and our proposed PKOOD model also achieves the best results in this dataset. However, the gap between ERM and other baseline models is more significant than that observed in the Adult dataset, suggesting that the OOD problem

Table 2: TopK Accuracy result in NICO++ dataset.

| Method | Acc@1 | Acc@3 | Acc@5 | Acc@10 |
|---|---|---|---|---|
| ERM | 66.3800 | 82.0730 | 86.9957 | 92.5601 |
| IRMv1 | 68.4692 | 84.8175 | 89.6256 | 94.5043 |
| GroupDRO | 67.2660 | 84.2864 | 89.1820 | 94.4079 |
| CVarDRO | 67.7045 | 83.2694 | 87.9491 | 93.0469 |
| StableNet | 68.7709 | 84.2236 | 88.6113 | 93.4158 |
| MLDG | 68.9108 | 84.9266 | 89.8245 | 94.4338 |
| MAPLE | 68.3835 | 83.6865 | 88.5085 | 93.4454 |
| PKOOD | **70.1809** | **85.8537** | **90.3502** | **94.9917** |

is more pronounced in the NICO++ dataset. Notably, the relative improvement of our model is larger in this dataset, which demonstrates the robustness of our model to the OOD problem.

## 5.4 ABLATION ANALYSIS

In this section, ablation analyses are conducted to validate the effectiveness of different parts of the PKOOD model. For simplicity, only the smaller Adult dataset with the maximum degree is used here.

Firstly, the usefulness of the $bias_{known}$ regularizer $\mathcal{L}_k$ and the $bias_{unknown}$ regularizer $\mathcal{L}_u$ are investigated. To achieve this, three variants of the PKOOD model are proposed, $\text{PKOOD}(\lambda_1 = 0)$, $\text{PKOOD}(\lambda_2 = 0)$ and $\text{PKOOD}(\lambda_3 = 0)$, by setting the hyperparameters $\lambda_1, \lambda_2, \lambda_3$ as 0, respectively. The results are shown in Figure 3 Left. It can be seen from the results that all variants perform worse than the original PKOOD model, which highlights the necessity of reducing the bias of both known and unknown. Furthermore, solving the known distribution shift (controlled by $\lambda_1, \lambda_2$) is more important than the unknown distribution shift (controlled by $\lambda_3$) on this dataset since removing the related regularize leads to a greater reduction in performance. This demonstrates the importance of addressing the partially known OOD problem.

Next, we validate the impact of meta-learning. We obtain a variant reweight-fix to remove meta-learning by inferring domain weights $\alpha$ from PKOOD and fixed as the sample weights to train a new simple reweight methods to get model parameter $\Theta$. Its performance is shown in Figure 3 Right. We also compare to Reweight-0.6 with $\delta = 0.6$, which is the best reweight method in the case study. Our PKOOD outperforms the reweight-fix, indicating that our meta-learning training framework not only obtains good domain weights $\alpha$ but also achieves model parameters $\Theta$ that generalize well to OOD data by simulating distribution shift. It is also observed that the reweight-fix achieves better performance than the reweight-0.6, demonstrating our PKOOD is able to learn good domain weights.

## 6 CONCLUSION

In this paper, we address the issue of partially known out-of-distribution generalization. Our approach involves a bilevel meta-learning framework that allows us to simulate the known distribution shift and automatically determine an effective reweighting of the training samples to achieve robust generalization performance on unknown test datasets. Besides, we derive an upper bound of the risk gap between the reweighted training samples and the target dataset to guide the design of the loss.

**Limitations and Future work**. One potential limitation of our PKOOD is that it primarily focuses on reweighting at the domain level, which may not fully capture the nuances of distributional differences. Future work will aim to expand it by incorporating reweighting strategies at the sample level.

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
