# A APPENDIX

## A.1 THE FRAMEWORK OF PKOOD

The whole framework of our PKOOD is shown as follows:

---

**Algorithm 1** Training Algorithm for *Meta Domain Reweighting for Partially Known Out-of-Distribution Generalization* (PKOOD)

---

**Require:** the training data $\mathcal{D}_s$, the distribution $\beta$ of known domain $\mathcal{D}_i$ on training data and the distribution $\gamma$ of known domain $\mathcal{D}_i'$ on test data.
**Ensure:** The domain weight $\alpha$ and the updated parameters $\Theta$.
1: Initialize all parameters.
2: **for** epoch $\leftarrow 0, 1, ..., total\_epoch\_number$ **do**
3:     **for** batch in all training dataset **do**
4:         Split mini-batch data into support/query sets.
5:         **Meta-train**: Calculate the support loss $\mathcal{F}(\alpha, \Theta)$ in Equation 7 on the support set, and one-step adapted parameters by gradient descent $\Theta' = \Theta - \lambda \frac{\partial \mathcal{F}(\alpha, \Theta)}{\partial \Theta}$.
6:         **Meta-test**: Calculate the query loss $\mathcal{G}(\Theta')$ in Equation 8 on the query set with the adapted parameters $\Theta'$.
7:         Update weight parameters $\alpha_{new} \leftarrow \lambda \frac{\partial \mathcal{G}(\Theta')}{\partial \alpha}$.
8:         Update model parameters $\Theta_{new} \leftarrow \lambda \frac{\partial (\mathcal{G}(\Theta') + \mathcal{F}(\alpha_{new}, \Theta))}{\partial \Theta}$.
9:     **end for**
10: **end for**

---

## A.2 DATASET STATISTICS

Two public datasets are used to evaluate the performance. The Adult is obtained from https://archive.ics.uci.edu/ml/datasets/adult and the NICO++ is from https://nicochallenge.com/. The detailed statistics of the datasets are shown in Table 3.

Table 3: The detailed statistics of the datasets.

| Dataset | Instances | Known Domains | Unknown Domains |
|---------|-----------|---------------|-----------------|
| Adult   | 48842     | 5             | 2               |
| NICO++  | 88866     | 4             | 6               |

To support our partial OOD setting, it is necessary to generate datasets with both a known and an unknown distribution shift. Therefore, the performance of our paper cannot be compareable to that of other papers using these datasets. The known domain for the Adult dataset is determined by the race feature, which includes categories such as 'White,' 'Asian-Pac-Islander,' 'Amer-Indian-Eskimo,' 'Black,' and 'Other'. The unknown domain of the Adult dataset is based on the sex feature, which includes the categories 'male' and 'female'. The unknown domain of Adult is based on sex feature, including 'male' and 'female'. The known domain of NICO++ is based on main classes, including 'transportation', 'animal', 'plant', and 'thing'. The main class, which is present in the original dataset, is marked by the class name. The unknown domain of NICO++ is based on context features, including 'dim', 'grass', 'outdoor', 'rock', 'water' and 'autumn'. The domain distribution of all datasets is shown in Table 4. The distance is the Euclidean distance between $\beta$ and $\gamma$.

Table 4: The known domain distribution of all datasets.

| Dataset | $\beta$ | $\gamma$ | Distance |
|---------|---------|----------|----------|
| Adult_5   | [0.387, 0.076, 0.230, 0.229, 0.076] | [0.075, 0.377, 0.076, 0.237, 0.232] | 0.486 |
| Adult_10  | [0.235, 0.046, 0.239, 0.435, 0.043] | [0.042, 0.246, 0.235, 0.041, 0.433] | 0.621 |
| Adult_20  | [0.463, 0.024, 0.023, 0.243, 0.244] | [0.022, 0.247, 0.461, 0.244, 0.024] | 0.696 |
| Adult_50  | [0.484, 0.009, 0.247, 0.249, 0.009] | [0.008, 0.245, 0.009, 0.243, 0.492] | 0.756 |
| Adult_100 | [0.254, 0.488, 0.004, 0.005, 0.247] | [0.250, 0.005, 0.244, 0.493, 0.006] | 0.767 |
| NICO++    | [0.581,0.279,0.115,0.022] | [0.240,0.129,0.252,0.377] | 0.533 |

## A.3 MODEL IMPLEMENTATION

All neural network approaches used in this study, including our proposed PKOOD, are implemented with PyTorch. We utilized baselines in their publicly available code, which are as follows:

- IRMv1:https://github.com/facebookresearch/InvariantRiskMinimization
- GroupDRO: https://github.com/kohpangwei/group_DRO
- CVaR DRO: https://github.com/daniellevy/fast-dro/
- StableNet: https://github.com/xxgege/StableNet
- MLDG: https://github.com/HAHA-DL/MLDG.git
- MAPLE: https://github.com/x-zho14/MAPLE

All the training was conducted on a single machine with an Intel(R) Xeon(R) Platinum 8163 CPU @ 2.50GHz and 8 Tesla V100 GPUs.

## A.4 THE PROOF

### A.4.1 THE PROOF OF THEOREM 1

To prove this theorem, we give two lemmas as follows:

**Lemma 1** (Hoeffding's Inequality). *If $X_1, X_2, ..., X_n$ are independent random variables with $a_i \leq X_i \leq b_i$ for all i, then for any $\epsilon > 0$,*

$$P[|\bar{X} - E[\bar{X}]| \geq \epsilon] \leq 2e^{-2n^2\epsilon^2/\sum_{i=1}^{n}(b_i-a_i)^2}, \tag{10}$$

*where $\bar{X} = (X_1 + ... + X_n)/n$.*

*Proof.* See Hoeffding (1994). □

**Lemma 2.** *For each $i \in \{1, 2, ..., N\}$, let $S_i$ be a labeled sample of size $\beta_i m$ generated by drawing $\beta_i m$ points from $\mathcal{D}_i$ and labeling them according to $f_i$. The number of samples in the sampled dataset is $\beta_i m$ where $m$ is the total number of samples. For any fixed weight vector $\alpha = (\alpha_1, \alpha_2, ..., \alpha_N)$, the ERM risk of the $\alpha$-weighted distribution on this samples is represented by $\hat{\epsilon}(\alpha)$, while the ideal risk is denoted by $\epsilon(\alpha)$. For any $\varepsilon > 0$ the following holds:*

$$P\left(|\hat{\epsilon}_\alpha(h) - \epsilon_\alpha(h)| \geq \varepsilon\right) \leq 2\exp\left(\frac{-2m\varepsilon^2}{\sum_{i=1}^{N}\frac{\alpha_i^2}{\beta_i}}\right) \tag{11}$$

*Proof.* (refer to Ben-David et al. (2010)). Let $X_{i,1}, X_{i,2}, ..., X_{i,\beta_i m}$ be random variables that take on the values $\frac{\alpha_i}{\beta_i}|h(x) - f_i(x)|$ for the $\beta_i m$ instances $x \in S_j$. Note that $0 \leq h(x) \leq 1$ and $0 \leq f_i(x) \leq 1$, then $\frac{\alpha_i}{\beta_i}|h(x) - f_i(x)| \in [0, \frac{\alpha_i}{\beta_i}]$ and thus $X_{i,1}, X_{i,2}, ..., X_{i,\beta_i m} \in [0, \frac{\alpha_i}{\beta_i}]$. Then

$$\hat{\epsilon}_\alpha(h) = \frac{1}{m}\sum_{i=1}^{N}\frac{\alpha_i}{\beta_i}\sum_{x \in S_i}|h(x) - f_i(x)| = \frac{1}{m}\sum_{i=1}^{N}\sum_{j=1}^{\beta_i m}X_{ij}$$

With $E[\hat{\epsilon}_\alpha(h)] = \epsilon_\alpha(h)$ and by Hoeffding's Inequality in Lemma 1, for any $\varepsilon > 0$,

$$P\left(|\hat{\epsilon}_\alpha(h) - \epsilon_\alpha(h)| \geq \varepsilon\right) \leq 2\exp\left(\frac{-2m^2\varepsilon^2}{\sum_{i=1}^{N}\sum_{j=1}^{\beta_i m}\text{range}^2(X_{ij})}\right) = 2\exp\left(\frac{-2m\varepsilon^2}{\sum_{i=1}^{N}\frac{\alpha_i^2}{\beta_i}}\right)$$

□

Then by this lemma, we prove the theorem 1.

**Theorem 1.** *Let $\mathcal{H}$ be a finite hypothesis space on $\mathcal{X}$. For any $\eta \in (0, 1)$, with probability at least $1 - \eta$,*

$$|\hat{\epsilon}_\alpha(h) - \epsilon_\alpha(h)| \leq \sqrt{\frac{1}{2m}\sum_{i=1}^{N}\frac{\alpha_i^2}{\beta_i}\ln(\frac{2|\mathcal{H}|}{\eta})}. \tag{12}$$

*Proof.* Let $\varepsilon = \sqrt{\frac{1}{2m}\sum_{i=1}^{N}\frac{\alpha_i^2}{\beta_i}\ln(\frac{2|\mathcal{H}|}{\eta})}$ where $|\mathcal{H}|$ is the size of the hypothesis space, then $2\exp\big(\frac{-2m\varepsilon^2}{\sum_{i=1}^{N}\frac{\alpha_i^2}{\beta_i}}\big) = \eta/|\mathcal{H}|$. And

$$P(|\hat{\epsilon}_\alpha(h) - \epsilon_\alpha(h)| > \varepsilon) \leq \sum_{h\in\mathcal{H}} P(|\hat{\epsilon}_\alpha(h) - \epsilon_\alpha(h)| \geq \varepsilon)$$

$$\leq \sum_{h\in\mathcal{H}} 2\exp\big(\frac{-2m\varepsilon^2}{\sum_{i=1}^{N}\frac{\alpha_i^2}{\beta_i}}\big) \quad \text{(by Lemma 2)}$$

$$= \sum_{h\in\mathcal{H}} \eta/|\mathcal{H}| \leq \eta$$

So for any $\eta \in (0,1)$, $|\hat{\epsilon}_\alpha(h) - \epsilon_\alpha(h)| \leq \sqrt{\frac{1}{2m}\sum_{i=1}^{N}\frac{\alpha_i^2}{\beta_i}\ln(\frac{2|\mathcal{H}|}{\eta})}$ with probability at least $1 - \eta$. $\qquad\square$

### A.4.2 THE PROOF OF THEOREM 2

**Theorem 2.** *Let $h^* = \underset{h\in\mathcal{H}}{\arg\min}\{\epsilon_\alpha(h) + \overline{\epsilon_t}(h)\}$ and $d_{\mathcal{H}\Delta\mathcal{H}}(\mathcal{D}_i, \mathcal{D}_j)$ is $\mathcal{H}$-divergence between domain $\mathcal{D}_i$ and $\mathcal{D}_j$, we have*

$$|\epsilon_\alpha(h) - \overline{\epsilon_t}(h)| \leq \epsilon_\alpha(h^*) + \overline{\epsilon_t}(h^*) + \frac{1}{2}\sum_{i=1}^{N}\sum_{j=i+1}^{N} |(\alpha_i\gamma_j - \alpha_j\gamma_i)| * d_{\mathcal{H}\Delta\mathcal{H}}(\mathcal{D}_i, \mathcal{D}_j). \tag{13}$$

*Proof.* Let $\lambda = \epsilon_\alpha(h^*) + \overline{\epsilon_t}(h^*)$ and $\epsilon_\alpha(h, h^*) = E_{x\sim\mathcal{D}_\alpha}[\mathcal{L}(h(x), h^*(x))]$. Then $\epsilon_\alpha(h) = E_{x\sim\mathcal{D}_\alpha}[\mathcal{L}(h(x), y)] = \epsilon_\alpha(h, y)$. Suppose that $\mathcal{L}$ is a distance function in regression such as Mean absolute Error (MAE) or Mean Squared Error (MSE), then $\epsilon$ satisfies the triangle inequality with $\epsilon_\alpha(h^*) + \epsilon_\alpha(h, h^*) \geq \epsilon_\alpha(h)$ and $\epsilon_\alpha(h^*) + \epsilon_\alpha(h^*) \geq \epsilon_\alpha(h, h^*)$. So

$$|\epsilon_\alpha(h) - \epsilon_\alpha(h, h^*)| \leq \epsilon_\alpha(h^*) \tag{14}$$

We have

$$|\epsilon_\alpha(h) - \overline{\epsilon_t}(h)| = |\epsilon_\alpha(h) - \epsilon_\alpha(h, h^*) + \epsilon_\alpha(h, h^*) - \overline{\epsilon_t}(h, h^*) + \overline{\epsilon_t}(h, h^*) - \overline{\epsilon_t}(h)|$$

$$\leq |\epsilon_\alpha(h) - \epsilon_\alpha(h, h^*)| + |\epsilon_\alpha(h, h^*) - \overline{\epsilon_t}(h, h^*)| + |\overline{\epsilon_t}(h, h^*) - \overline{\epsilon_t}(h)|$$

$$\leq \epsilon_\alpha(h^*) + |\epsilon_\alpha(h, h^*) - \overline{\epsilon_t}(h, h^*)| + \overline{\epsilon_t}(h^*) \quad \text{(by Equation 14)}$$

$$\leq \epsilon_\alpha(h^*) + \overline{\epsilon_t}(h^*) + \frac{1}{2}d_{\mathcal{H}\Delta\mathcal{H}}(\mathcal{D}_\alpha, \overline{\mathcal{D}_t}) \quad \text{(by definition of } \mathcal{H} \text{ divergence)}$$

$$= \lambda + \frac{1}{2}d_{\mathcal{H}\Delta\mathcal{H}}(\mathcal{D}_\alpha, \overline{\mathcal{D}_t})$$

$$= \lambda + \sup_{h,h'\in\mathcal{H}} |\mathrm{Pr}_{x\sim\mathcal{D}_\alpha}[h(x) \neq h'(x)] - \mathrm{Pr}_{x\sim\overline{\mathcal{D}_t}}[h(x) \neq h'(x)]|$$

$$= \lambda + \sup_{h,h'\in\mathcal{H}} |\mathbb{E}_{x\sim\mathcal{D}_\alpha}[|h(x) - h'(x)|] - \mathbb{E}_{x\sim\overline{\mathcal{D}_t}}[|h(x) - h'(x)|]|$$

$$= \lambda + \sup_{h,h'\in\mathcal{H}} |\sum_{i=1}^{N}\alpha_i\mathbb{E}_{x\sim\mathcal{D}_i}[|h(x) - h'(x)|] - \sum_{j=1}^{N}\gamma_j\mathbb{E}_{x\sim\mathcal{D}_j}[|h(x) - h'(x)|]|$$

$$= \lambda + \sup_{h,h'\in\mathcal{H}} |\sum_{i=1}^{N}(\sum_{j=1}^{N}\gamma_j)\alpha_i\mathbb{E}_{x\sim\mathcal{D}_i}[|h(x) - h'(x)|] - \sum_{j=1}^{N}(\sum_{i=1}^{N}\alpha_i)\gamma_j\mathbb{E}_{x\sim\mathcal{D}_j}[|h(x) - h'(x)|]|$$

$$= \lambda + \sup_{h,h'\in\mathcal{H}} |\sum_{i=1}^{N}\sum_{j=1}^{N}\alpha_i\gamma_j\big(\mathbb{E}_{x\sim\mathcal{D}_i}[|h(x) - h'(x)|] - \mathbb{E}_{x\sim\mathcal{D}_j}[|h(x) - h'(x)|]\big)|$$

$$= \lambda + \sup_{h,h'\in\mathcal{H}} |\sum_{i=1}^{N}\sum_{j=i+1}^{N}(\alpha_i\gamma_j - \alpha_j\gamma_i)\big(\mathbb{E}_{x\sim\mathcal{D}_i}[|h(x) - h'(x)|] - \mathbb{E}_{x\sim\mathcal{D}_j}[|h(x) - h'(x)|]\big)|$$

$$\leq \lambda + \sum_{i=1}^{N}\sum_{j=i+1}^{N}|(\alpha_i\gamma_j - \alpha_j\gamma_i)| * 2\sup_{h,h'\in\mathcal{H}}|\mathbb{E}_{x\sim\mathcal{D}_i}[|h(x) - h'(x)|] - \mathbb{E}_{x\sim\mathcal{D}_j}[|h(x) - h'(x)|]$$

$$= \lambda + \sum_{i=1}^{N}\sum_{j=i+1}^{N}|(\alpha_i\gamma_j - \alpha_j\gamma_i)| * d_{\mathcal{H}\Delta\mathcal{H}}(\mathcal{D}_i, \mathcal{D}_j).$$

$\square$

### A.4.3 THE PROOF OF THEOREM 3

**Theorem 3.** *The upper bound of $|\bar{\epsilon}_t(h) - \epsilon_t(h)|$ is irrelevant to $\alpha$ and under the assumption of IRM invariant predictor* Chang et al. (2020) *over unknown domains $\mathcal{G}_2$, its expectation is zero.*

*Proof.* The assumption of IRM is that features $X$ can be split into invariant feature $X_v$ and spurious feature $X_s$ and $p(Y|X_v)$ is stable across each environment Arjovsky et al. (2020). Then for split sub-domains $\mathcal{D}_i$ and $\mathcal{D}_i'$, we have $\Pr_{x \sim \mathcal{D}_i}(Y|X_v) = \Pr_{x \sim \mathcal{D}_i'}(Y'|X_v)$. Let $h = w(\Phi(x))$, the goal of IRM is

$$h_{un}^* = w^* \circ \Phi^*$$

$$= \underset{\Phi:\mathcal{X}\to\mathcal{H}, w:\mathcal{H}\to\mathcal{Y}}{arg\,min} \sum_{e \in \mathcal{E}_{tr}} \epsilon_e(\Phi \circ w)$$

$$subject\ to\ w \in \underset{\overline{w}:\mathcal{H}\to\mathcal{Y}}{arg\,min}\ \epsilon_e(\Phi \circ \overline{w}), for\ all\ e \in \mathcal{E}_{tr}$$

And

$$|\bar{\epsilon}_t(h) - \epsilon_t(h)|$$

$$= |\sum_{i=1}^{N} \gamma_i \epsilon_i(h) - \sum_{i=1}^{N} \gamma_i \epsilon_i'(h)|$$

$$= |\sum_{i=1}^{N} \gamma_i(\epsilon_i(h) - \epsilon_i'(h))|.$$

We also have

$$\epsilon_i(h_{un}^*) - \epsilon_i'(h_{un}^*) = \mathrm{E}_{x \sim \mathcal{D}_i}(|h_{un}^*(x) - f_i|) - \mathrm{E}_{x \sim \mathcal{D}_i'}(|h_{un}^*(x) - f_i'|)$$

$$= \Pr_{x \sim \mathcal{D}_i}(|h_{un}^*(x) \neq f_i|) - \Pr_{x \sim \mathcal{D}_i'}(|h_{un}^*(x) \neq f_i'|)$$

$$= \Pr_{x \sim \mathcal{D}_i}(|w^*(\Phi^*(x)) \neq f_i|) - \Pr_{x \sim \mathcal{D}_i'}(|w^*(\Phi^*(x)) \neq f_i'|)$$

$$= -(\Pr_{x \sim \mathcal{D}_i}(Y = f_i|(\Phi^*(x))) - \Pr_{x \sim \mathcal{D}_i'}(Y' = f_i'|(\Phi^*(x)))).$$

Since $\Pr_{x \sim \mathcal{D}_i}(Y = f_i|(\Phi^*(x))) = \Pr_{x \sim \mathcal{D}_i'}(Y' = f_i'|(\Phi^*(x)))$ and $w^*$ is optimal for all environments, $\mathrm{E}(\epsilon_i(h_{un}^*) - \epsilon_i'(h_{un}^*)) = 0$ and it leads to $E[|\bar{\epsilon}_t(h_{un}^*) - \epsilon_t(h_{un}^*)|] = 0$. □

### A.4.4 THE PROOF OF COROLLARY 1

**Corollary 1.** *For $\forall i \in \{1, ..., N\}, \alpha_i = \beta_i$, the upper bound of variance in Theorem 1 gets minimum value.*

*Proof.* By Cauchy–Schwarz Inequality, we have $\sum_{i=1}^{N} \frac{\alpha_i^2}{\beta_i} = (\sum_{i=1}^{N} \frac{\alpha_i^2}{\beta_i})(\sum_{j=1}^{N} \beta_j) \geq (\sum_{i=1}^{N} \frac{\alpha_i}{\sqrt{\beta_i}} * \sqrt{\beta_i})^2 = 1$ where the equation is achieved when $\frac{\alpha_i^2}{\beta_i} = \beta_i$, i.e., $\alpha_i = \beta_i$. □

### A.4.5 THE PROOF OF COROLLARY 2

**Corollary 2.** *For $\forall i \in \{1, ..., N\}, \alpha_i = \gamma_i$, the upper bound of $bias_{known}$ in Theorem 2 gets minimum value.*

*Proof.* Following the work of Ben-David et al. (2010), we assume the existence of a hypothesis $h^* = \underset{h \in \mathcal{H}}{arg\,min}\{\epsilon_\alpha(h) + \bar{\epsilon}_t(h)\}$ with low error on both the data with reweighted $\alpha$ distribution and the target domain. If such a hypothesis does not exist, the difference between train and test distributions would be too significant, making it impossible for any OOD method to overcome the problem. Therefore the term $\epsilon_\alpha(h^*) + \bar{\epsilon}_t(h^*)$ becomes much smaller than the next term $\frac{1}{2}\sum_{i=1}^{N}\sum_{j=i+1}^{N}|(\alpha_i\gamma_j - \alpha_j\gamma_i)| * d_{\mathcal{H}\Delta\mathcal{H}}(\mathcal{D}_i, \mathcal{D}_j)$ in the upper bound. $|(\alpha_i\gamma_j - \alpha_j\gamma_i)|$ achieves minimum value 0 when $\alpha_i = \gamma_i$. □

## A.5 THE CONVERGENCE OF $\alpha$

In this section, we analyze the behavior of the learned domain weights $\alpha$ during the training process. For clarity, we scale $\alpha$ into the range of [0, 1]. The normalized $\alpha$ is calculated using the formula $\frac{\alpha-\beta}{\gamma-\beta}$. When $\alpha = \beta$, the normalized $\alpha$ equals 0 and when $\alpha = \gamma$, the normalized $\alpha$ is 1. The curve depicting the variation of normalized $\alpha$ for different known domains across epochs is presented in Figure 4. It can be observed that the $\alpha$ eventually converges. Furthermore, the weights assigned to the different known domains are not identical, thereby confirming the conclusions drawn from previous theorems and corollaries.

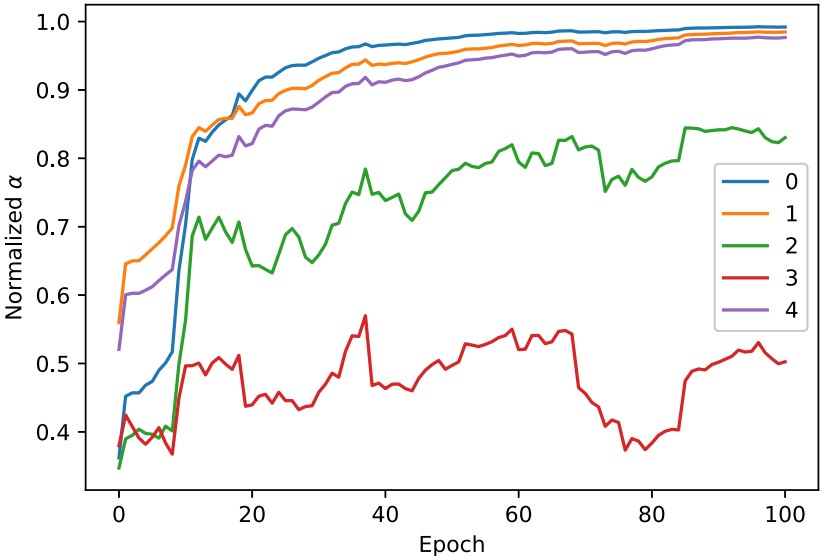

Figure 4: The curve of normalized $\alpha$ of different known domains with respect to the epoch.

## A.6   ONLINE A/B TEST

To further validate the effectiveness of the proposed method, we deployed PKOOD in a real-world budget-constrained coupon recommendation system application at the beginning of a promotional period. We performed an online A/B test to assess the performance of our system against base models.

During the promotional period, coupon distribution is mainly determined by contractual agreements with merchants, which typically specify the exposure ratio of a particular coupon type (usually belonging to a specific business unit). This differs significantly from the pre-promotion distribution. Additionally, training data is collected before the promotion, making it challenging to optimize target performance (i.e., the conversion rate or CVR) during the promotional period. However, we can estimate the coupon distribution during the promotion based on contractual agreements, while user distribution remains unknown. This problem is consistent with our proposed partial OOD setting, and we can leverage PKOOD to enhance our performance.

To simplify the description, we grouped similar business units (BUs) into a single domain, indexed by [0-9]. The distribution of domains before and during the promotion period, denoted by $\beta$ and $\gamma$ respectively, is shown in Table 5.

Table 5: domain distribution before and during the promotion

| domain | 0 | 1 | 2 | 3 | 4 | 5 | 6 | 7 | 8 | 9 |
|---|---|---|---|---|---|---|---|---|---|---|
| $\beta$ | 0.099 | 0.102 | 0.087 | 0.107 | 0.104 | 0.082 | 0.119 | 0.085 | 0.096 | 0.118 |
| $\gamma$ | 0.718 | 0.225 | 0.030 | 0.014 | 0.006 | 0.001 | 0.002 | 0.001 | 0.001 | 0.002 |

As shown in Table 5, the coupon distribution before the promotion was more uniform, whereas during the promotion, a few key BUs covered the majority of the exposure.

Table 6: The online A/B test results

| method | CVR |
|---|---|
| ERM | - |
| MAPLE | 0.28% ↑ |
| PKOOD | 0.46% ↑ |

Since MAPLE is one of the best methods among the baselines in previous experiments, we compared PKOOD with ERM and MAPLE in this context. The online A/B test results are shown in Table 6. Compared to ERM,

MAPLE achieved a relative improvement of 0.28% in CVR, while PKOOD achieved a relative improvement of 0.46%. These results highlight the significance of addressing the OOD issue and conclusively demonstrate that PKOOD can reap more substantial practical benefits than the base models. Following this, we deployed our model online.