# OpenReview forum: "Meta Domain Reweighting for Partially Known Out-of-Distribution Generalization"
_ICLR.cc/2024/Conference — ICLR 2024 Conference Withdrawn Submission_

### Official Review · Reviewer_3iTa · 2023-10-30

**Soundness:** 3 good
**Presentation:** 3 good
**Contribution:** 2 fair
**Rating:** 5
**Confidence:** 3

**Summary:**

This paper considers a new domain generalization setting, called Partially Known Out-of-Distribution Generalization (PKOOD), where test data is unavailable during training stage but some prior knowledge (e.g., the data proportion of different environments or concepts) of test domain is accessible during training. Accordingly, the distribution shift is divided into two parts: known and unknown. For the PKOOD setting, the paper proposes a Meta Domain Reweighting approach to simulate the known distribution shift and conduct invariant learning via a bilevel meta-learning framework with theoretical support. Experiments on two datasets (Adult and NICO++) are conducted to validate the effectiveness of the proposed approach.

**Strengths:**

1. The method is supported with theoretical analysis.
2. The paper is well-structured and well-written.
3. The experiments consistently show the effectiveness of the proposed method.

**Weaknesses:**

1. Although authors propose a new DG setting, called PKOOD, its significance in practice is not quite clear to me. Especially, the known prior knowledge of test domain in the paper seems to be in the form of data proportions of certain concepts, while this may be a somewhat strict prior knowledge that needs to know all test data beforehand.
2. The batch size used in the experiments is 1024. Does the proposed method require large batch size to work well?
3. The proposed method is relatively complex. It is better to provide the algorithm and complexity analysis of the method.
4. Authors claim that the bilevel meta domain reweighting method can simulate the known distribution shift and achieve invariant learning. Is there any qualitative experiment to verify these, e.g., the distance between learned $\alpha$ and known $\gamma$ and the t-SNE visualization of training domain and testing domain?
5. Why does Fig. 2(c) show completely different trend of $\delta$ from Fig. 2(a) and (b)? Could you please explain this?

**Questions:**

Please refer to the aforementioned weaknesses. Besides, discussion about the ethics is needed, since the construction of Adult dataset in PKOOD setting involves races and genders.

---

### Official Review · Reviewer_kPru · 2023-10-31

**Soundness:** 2 fair
**Presentation:** 2 fair
**Contribution:** 2 fair
**Rating:** 5
**Confidence:** 3

**Summary:**

The paper identifies a new practical problem setting: partially-known out-of-distribution generalization. In this setting the distribution of the domains in the target dataset is known, but not the allocation of the individual examples to the observed domains. There may also be other aspects of the domains that are not known (considering that an example may be categorized into multiple domains at the same time - e.g. based on race and gender as it is in the considered Adult dataset). The setting is realistic because the proportion of the considered domains can be identified by sampling a small number of examples or be enforced via interventions. A new meta-learning based method is proposed as part of the paper to target this newly identified problem, inspired by theoretical analysis.

**Strengths:**

* The paper identifies and establishes a new practically useful scenario relevant to domain generalization (DG): DG with partial knowledge - more specifically knowledge about the proportion of various domains in the target set.
* There is extensive theoretical analysis of the upper bound of the target empirical risk and it is used to motivate the proposed method.
* The proposed method leads to improvements compared to the other evaluated methods.
* There is an ablation study on the different components of the objective used.

**Weaknesses:**

* The method is evaluated on a small number of datasets: two. Using more scenarios would make the paper more persuasive in showing that the method really helps. Ideally at least one further dataset would be added, especially from the area of computer vision or NLP as these have been of primary interest in deep learning. A further tabular dataset would help strengthen the paper even further. Swapping the known and unknown domains in the two already considered datasets could also be interesting.
* It would be valuable to consider stronger DG baselines: e.g. on NICO++ the best approaches seem to be SWAD, EoA or CORAL but neither is included in this paper. This would be important in confirming if the proposed method is state-of-the-art for the identified setting.
* A measure of variability across runs would be good to add (e.g. stdevs or SEMs), to see how significant the improvements from the method are.
* The loss that is optimized includes a large number of components so it makes the method quite complex.
* The presentation could be improved: e.g. adding explanation to Figure 1 caption that only the proportions of the domains are known and not also the allocation of examples, explaining the practical implications of the theoretical analysis more and connecting it more strongly to the design of the method, fixing minor typos (e.g. “The regularizer of bias known is definded as follows:”, “Domain weights \alpha is updated based on the query loss.”).
* The ablation study would be more interesting if conducted on NICO++ as the Adult dataset can be seen as only a toy dataset with potentially limited more general implications.

**Questions:**

* How do stronger DG baselines compare?
* How were the hyperparameters selected for the proposed method and the baselines? (these were shown to be quite important for fair evaluation in DG)

Plus questions that follow from what is listed in the weaknesses.

---

### Official Review · Reviewer_2X6w · 2023-10-31

**Soundness:** 1 poor
**Presentation:** 2 fair
**Contribution:** 1 poor
**Rating:** 3
**Confidence:** 4

**Summary:**

This paper introduces a model-agnostic reweighting method entitled Meta Domain Reweighting for Partially Known Out-Of-Distribution Generalization (PKOOD). The approach exploits known facets of the distribution shift, to reweight training samples such that you achieve strong generalization for the testing domain. It is based on the assumption that some priors related to the causal factors of distribution shifts can be acquired.

The authors make a clear distinction between general Out-Of-Distribution (OOD) generalization, Domain Adaptation (DA), and PKOOD (see Fig. 1). For PKOOD, the paper assumes that you have two main directions of shift, one denoted as a known shift and the other as an unknown shift. The paper assumes that you can have a prior for one of the directions, while the other remains unknown. As shown in Figure 1, the known shift can refer to a change in the distribution of the semantic category, and the unknown shift can refer to a change in the distribution of the considered backgrounds. According to Sec. 5.1. (datasets NICO++ and Adult), the paper assumes subpopulation shifts for the known domains (no new domains during testing, only different proportions) and general covariate shifts for the unknown domains (new domains appear during testing).

Claimed contributions:
  - introducing the PKOOD setup
  - introducing the method: Meta Domain Reweighting for PKOOD, along with a theoretical justification of the approach
  - experimental validation of the proposed approach

**Strengths:**

**S1** Good experimental results when compared to other domain generalization approaches.

**Weaknesses:**

**W1** The theoretical justification of the approach.

Eq. 1 bounds the gap between target risk and the empirical risk $ \hat{\epsilon} {\alpha}(h) $, computed after an $\alpha$ re-weighting of the source data.
It remains unclear why the ideal $\alpha$-weighted error was included ($\epsilon_\alpha(h)$) as we are unable to estimate this in practice. The triangle inequality could have been reduced to include on the right-hand side only $bias_{unknown}$ and a $bias_{known}$ equal to $  | \hat{\epsilon} {\alpha}(h)- \overline{\epsilon} t(h)|$ .
This alteration would lead to the conclusion that the optimal $\alpha$ corresponds to $\gamma$, which makes sense if the target is the optimization of $\epsilon_t(h)$.
This ideal error is present both in $variance$ and $bias_{known}$ terms, but further when they are employed for regularizing the model, the $variance$ term is not included, while from the bound of the $bias_{known}$ we only consider terms that are not connected to this ideal estimation.

**W2** The proposed approach is not presented fairly in comparison to MAPLE [1] and MLDG [2], the methods on which it is built.

**W3** The experimental analysis should be improved - see questions **Q1**-**Q5**


 [1] Zhou et al. Model Agnostic Sample Reweighting for Out-of-Distribution Learning - ICML 2022

[2] Li et al. Learning to Generalize: Meta-Learning for Domain Generalization - AAAI 2018

**Questions:**

**Q1** Sec. 5.2.

In Figure 2 a), when comparing between $\delta=0$ and $\delta=1$, it seems to be better to consider solely $\beta$ than to solely consider $\gamma$. This is somehow counterintuitive for the general PKOOD setup.

*Q1.1* For this experiment, one of the unknown domains of Adult dataset is used for training and the other for testing? If yes, have you considered evaluating different $\delta$ values while keeping the unknown domain fixed between train and test? This would isolate the impact of the unknown domains and give a clean intuition for $\delta$.

*Q1.2* For this experiment, what training methodology have you considered? (as you are having a fixed re-weighting I assume there is a different strategy than the one proposed in the main algorithm and should be specified in the paper)

**Q2** Table 1 - evaluation on Adult dataset

*Q2.1* For this set, the $\mathcal{L}_u$ loss term is applied over a single unknown domain present during training?

*Q2.2* The approach combines the ideas of MLDG and MAPLE. Consequently,  when reporting the improvement, the focus should be MLDG and MAPLE and not ERM. Also, as the current approach employs IRMv1, you should specify the risk considered for MAPLE when reporting the results in Table 1.

*Q2.3* How do you apply IRMv1 for this set ? Which are the training environments?

*Q2.4* How do you explain the performance evolution w.r.t. the distance between $\beta$ and $\gamma$? For example, we have a performance peak for Adult_20, with ERM.
In Sec. 5.3. it is mentioned that results between different degrees are not comparable due to the randomly generated splits, yet the evolution between degrees is employed for concluding that the proposed approach is better than ERM for more severe shifts, so the order should be relevant.

**Q3** Ablation studies - Fig 3 Left

*Q3.1* Regarding the impact of loss components, which are the default values for $\lambda_1$, $\lambda_2$ and $\lambda_3$?

*Q3.2* PKOOD($\lambda_3=0$) shows the smallest decrease for the Adult set. Can you perform the same ablation on NICO++?
As Adult has only one unknown domain during training, it is not enough for evaluating the impact of $\mathcal{L}_{u}$

**Q4** Ablation studies - Fig 3 right

*Q4.1* Can you explain again how reweight-fix is implemented? You set $\alpha=\beta$? (this should also be clarified in the paper)
How do you correlate the results of Fig. 3 with results of Fig. 2 a)  (w.r.t. reweight-fix) ?

*Q4.2* The last sentence in Sec.5.4 mentions that "It is also observed that the reweight-fix achieves better performance than the reweight-0.6, demonstrating our PKOOD is able to learn good domain weights."
We cannot conclude this by comparing reweight-fix with reweight-0.6. Can you clarify this?

**Q5** Additional ablation studies are required

*Q5.1* There is no analysis regarding the impact of using $\gamma$ in $\mathcal{L}_k$, which is a crucial step in assessing the importance of the proposed PKOOD method.

*Q5.2* It would be relevant to see the results of the ablations on NICO++ dataset, as it captures a more complex scenario.

*Q5.3* To understand the impact of the PKOOD setup, you need to perform an experimental analysis where you separate the two shifts (known and unknown).

*Q5.4* You should consider a measure of the two shifts (e.g. OTDD distances between datasets [3]), to evaluate the impact of each loss component correctly. (e.g. $\mathcal{L}_u$ may be irrelevant for small shifts between unknown domains, but crucial for larger shifts)

[3] Alvarez-Melis and Fusi "Geometric dataset distances via optimal transport" -NeurIPS 2020

**Minor comments**:
   - Fig. 1 - percentage values are hard to read as they overlap with the bars behind (for train set & PKOOD scenarios)
  - Theorem 2 - I think $\frac{1}{2}$ should not appear in the formula - please check
  - Eq. 4 - the loss $l$ is between $h(x_j)$ and $y_j$ (check paranthesis)
  - Sec. 5.1. -  not clear if the mentioned embedding and batch sizes are for Adult or NICO++ (I assume Adult, but should be clearly specified)
  - Table 1 - top-left cell, with text "Method\nDistance" should be revised (e.g. the method is below, but method is written above Distance)

---

### Official Review · Reviewer_Wby4 · 2023-11-02

**Soundness:** 3 good
**Presentation:** 2 fair
**Contribution:** 2 fair
**Rating:** 3
**Confidence:** 4

**Summary:**

This paper studies a novel OOD generalization setting: Partially Known OOD Generalization, where the group distribution knowledge is known, and the categorical knowledge remains unknown. In order to leverage the known information and transfer the source knowledge to unknown target domains, the authors propose to generate the $\alpha$ weighted data from source distributions as a proxy to bridge the known source domain and partially known target domain. By conducting meta-learning to minimize an error bound, the $\alpha$ weight can be automatically optimized to minimize the variance and bias. As a result, the generalization of OOD data can be further improved. Through quantitative validation on two datasets: Adult and NICO++, the authors prove that the proposed method can achieve better performance than several baseline methods.

**Strengths:**

-  The proposed method is theoretically supported and proposed with reasonable logic.
- The writing is good; this paper is not hard to understand.
- Such problem setting is novel and interesting and could provide some insights to further OOD generalization research.

**Weaknesses:**

- First, I hope the author could provide more justification on why such a partially known OOD generalization is of great significance. This paper regards the group distribution as an additional information known for target domains, are there any other factors that could be helpful from the quote ‘some facets of the distribution shift can be predicted’?
- The methodology mainly borrows meta-learning and IRM to reduce variance and bias, respectively. Although it is a reasonable design, the contributions are quite limited.
- The experiments are not sufficiently conducted Only two datasets are used, many popular datasets such as the datasets in DomainBed and Wilds are not considered.
- The compared baseline methods are insufficient, most of the comparisons are not recent advances. Please considers some SOTA methods such as SWAD, Cha et al, Domain Generalization by Seeking Flat Minima; Sun et al., CORAL, Correlation Alignment for Unsupervised Domain Adaptation, etc.

**Questions:**

Please see the weaknesses for details.

**Details Of Ethics Concerns:**

No ethic concerns.